# Gold Nanoparticle Mesoporous Carbon Composite as Catalyst for Hydrogen Evolution Reaction

**DOI:** 10.3390/molecules29153707

**Published:** 2024-08-05

**Authors:** Erik Biehler, Qui Quach, Tarek M. Abdel-Fattah

**Affiliations:** Applied Research Center at Thomas Jefferson National Accelerator Facility, Department of Molecular Biology and Chemistry at Christopher Newport University, Newport News, VA 23606, USA; erik.biehler@cnu.edu (E.B.);

**Keywords:** nanocomposite, hydrogen evolution, mesoporous carbon, gold nanoparticles, mesoporous carbon, sustainable source

## Abstract

Increased environmental pollution and the shortage of the current fossil fuel energy supply has increased the demand for eco-friendly energy sources. Hydrogen energy has become a potential solution due to its availability and green combustion byproduct. Hydrogen feedstock materials like sodium borohydride (NaBH_4_) are promising sources of hydrogen; however, the rate at which the hydrogen is released during its reaction with water is slow and requires a stable catalyst. In this study, gold nanoparticles were deposited onto mesoporous carbon to form a nano-composite catalyst (AuNP-MCM), which was then characterized via transmission electron microscopy (TEM), powder X-ray diffraction (P-XRD), and scanning electron microscopy/energy dispersive X-ray spectroscopy (SEM/EDS). The composite’s catalytic ability in a hydrogen evolution reaction was tested under varying conditions, including NaBH_4_ concentration, pH, and temperature, and it showed an activation of energy of 30.0 kJ mol^−1^. It was determined that the optimal reaction conditions include high NaBH4 concentrations, lower pH, and higher temperatures. This catalyst, with its stability and competitively low activation energy, makes it a promising material for hydrogen generation.

## 1. Introduction

Hydrogen fuel is an alternative type of fuel that has potential in solving the world’s energy crisis [1]. The energy generation of gasoline combustion is lower than the energy released from hydrogen combustion [2]. When hydrogen is used as a fuel, the major byproduct, in terms of its energy reaction, is water, so its environmental impact is minimal [2]. Furthermore, hydrogen can be generated from the reaction between hydrogen feedstock, such as sodium borohydrides (NaBH_4_), and water [3], as seen in Equation (1). The widespread implementation of hydrogen as a fuel would reduce dependence on fossil fuels; however, the biggest disadvantage of this reaction is the slow rate of reaction between metal hydrides and water. As such, a catalyst is necessary before hydrogen gas can become a viable fuel source [3].
NaBH_4_ + 2H_2_O → 4H_2_ + NaBO_2_(1)

Among the catalysts, nanoparticles have been the focus of much scientific study due to their unique properties, including their catalytic properties [4,5]; however, it is difficult to control nanoparticle performance in reactions as this depends greatly on characteristics such as shape, size, crystal structure, and texture [6,7,8]. The agglomeration of nanoparticles often affects their size and structure and leads to the degeneration of catalytic ability [9,10]. In order to mitigate this issue, the nanoparticles can be imbedded on a carbon template to prevent their agglomeration and improve the durability of the material in catalytic reactions [5,11].

One family of durable carbon materials that also has catalytic potential is that of mesoporous carbon materials (MCMs) [12,13]. The different types of porous carbons are characterized by their differences in pore size, with mesoporous carbon materials having a pore size 2–50 nm, while microporous and macroporous carbon have smaller and larger pore sizes, respectively [14,15]. The narrow pore size range allows for significant control of various characteristics during synthesis, which includes thermal, mechanical, and electrical stability, chemical inertness, ordered pore structure, large surface area, pore volume, and catalytic activity [16,17]. Most traditional metal and biological catalysts have several disadvantages, such as high reaction temperature, long reaction time, low conversion ratio, and low regioselectivity, so research is being conducted with MCMs to reduce these disadvantages [18]. For example, factors like cost, complexity of synthesis, and potential wasted materials can be reduced by using MCMs as opposed to traditional catalysts as MCMs are only made of ordered carbon atoms that are synthesized via well-established procedures [19].

One such method of synthesizing MCMs involves filling mesoporous silica with a carbon precursor like sucrose, which undergoes several high-temperature processes. The silica is removed with hydrofluoric acid, which leaves only carbon in the form of MCMs [20]. Unfortunately, this process involves aggressive chemicals that, along with intermediary products, become dangerous waste products. One novel method of forming MCMs has recently been developed, which involves using starches and expanding the present pores to increase the catalytic potential [21]. Starches are typically high-density substances, which makes them catalytically inert; but by expanding the pores, this activity was increased [21]. The pore expansion temperature is the key to controlling pore size and volume, with a similar process used on plant fibers resulting in similar results [21,22]. The benefits of this new method include the starting material being completely nontoxic and renewable, the lack of toxic materials needed for the reaction, and the ease of synthesis [21,23,24].

For this study, MCMs were decorated with gold nanoparticles to form a nanocomposite catalyst (AuNP-MCM) that is highly active, stable, and easily recyclable. The synthesized AuNP-MCM was characterized via TEM, P-XRD, and SEM-EDS. The catalytic ability of nanocomposite was tested in the hydrogen evolution reaction under various pHs (6, 7, 8), reactant concentrations (793 µmol, 952 µmol, 1057 µmol), and temperatures (273 K, 288 K, 295 K, 303 K). The recyclability of AuNP-MCM was also examined in reusability trials.

## 2. Results/Discussions

### 2.1. Catalyst Characterization

The gold nanoparticles for the AuNP-MCM seen in the TEM micrograph of Figure 1 vary greatly in diameter, but the particle specifically observed in Figure 1d under a smaller scale has a diameter of 20 nm. The SEM/EDS analysis in Figure 2a,b shows the gold nanoparticles on the mesoporous carbon with the gold concentration of 8.54% and carbon concentration of 10.73%. Figure 1b and Figure 2a also show that the gold nanoparticles are evenly distributed on the MCM backbone. 

The P-XRD spectrum seen in Figure 3 for the MCM showed a peak in the 20–30° range, correlating to the carbon framework, and a peak at the 40–45° range that also corresponds to the mesoporous carbon [25]. After AuNPs were deposited on the MCM, the observed peaks of the MCM were slightly shifted, but they were still within the acceptable range. The peaks at 40–45° disappeared in AuNP-MCM; the same phenomena were reported in some previous studies [26,27]. The peaks at 38°, 44°, and 64° corresponded with the (111), (200), and (220) lattice planes of gold nanoparticles [28]. 

### 2.2. Catalytic Tests

Figure 4 shows the catalytic ability of AuNP-MCM at different reactant concentrations. The catalyzed reaction achieved the highest hydrogen generated rate of 0.0346 mL min^−1^ mg^−1^ at 1057 µmol, while the reaction achieved the lowest rate of 0.0140 mL min^−1^ mg^−1^ at 793 µmol. At the concentration of 952 µmol, the reaction rate was 0.0159 mL min^−1^ mg^−1^. The hydrogen generated rate increased as the concentration of NaBH_4_ increased. The increase in the reactant concentration shifted the equilibrium of Equation (1) and led to the formation of more products [3,29]. 

The effect of pH on the catalyzed reactions was indicated in Figure 5. The reaction rate at pH 6 (0.0447 mL min^−1^ mg^−1^) was higher than that of pH 7 (0.0159 mL min^−1^ mg^−1^). The lowest reaction rate was 0.0087 mL min^−1^ mg^−1^ at pH 8. It had been stated in a previous kinetic study of sodium borohydride hydrolysis that the rate becomes lower as the pH increase due to the inhibition effect of OH^−^ [29]. The free proton at low pH condition accelerated the hydrolysis process [29]. 

Figure 1 shows a proposed mechanism for the overall reaction, which could explain these results. NaBH4 reduces when hydrolyzed and forms a reversable complex with the metal nanoparticle at a catalytic site. An adjacent catalytic site receives a hydride ion and stabilizes the nanoparticle–borohydride complex, which reacts with the water to release hydrogen gas. This process repeats until there are no longer any hydride ions available and the complex breaks, releasing tetrahydroxylborate and H_2_ gas. More acidic pHs increase the concentration of H^+^ ions, which increase the conversion rate and supply of hydroxide ions for the conversion. 

Figure 6 shows that the reaction rate of catalyzed hydrogen evolution reaction improved at higher temperature. The hydrogen generation rates were 0.0062 mL min^−1^ mg^−1^, 0.0117 mL min^−1^ mg^−1^, 0.0159 mL min^−1^ mg^−1^, and 0.0232 mL min^−1^ mg^−1^ at 273 K, 288 K, 295 K, and 303 K, respectively. The same pattern was observed in other studies [3,4,29].

From the Arrhenius plot seen in Figure 7 and Appendix A, the activation energy for the AuNP-MCM catalyst is determined to be 30 kJ mol^−1^. The activation energy is also compared to previously recorded activation energies for other catalysts in Table 1, which shows it as one of the higher activation energies among other inorganic catalysts. AuNP-MCM has higher activation energy than other catalysts, except the gold nanoparticle supported over multiwalled carbon nanotubes (Au/MWCNTs). When compared to unsupported gold nanoparticles (BCD-AuNPs), there was a marked improvement. This indicates that the addition of MCM improves the catalytic ability of gold nanoparticles. These results highly imply that AuNP-MCM is an efficient catalyst for sodium borohydride hydrolysis.

### 2.3. Catalytic Reusability Tests

The AuNP-MCM catalyst underwent reusability trials to determine its reusability after five consecutive uses under conditions involving 952 µmol of NaBH_4_, a pH of 7, and at 295 K. The plot in Figure 8 shows that the catalytic activity for the AuNP-MCM catalyst drastically decreased in catalytic activity after the second trial. From the third trial, the catalytic ability was increased and remained consistent in the fourth and fifth trials. This indicates that if more trials were to have been conducted, the catalytic activity may eventually remain consistent, resulting in a more ideal and durable catalyst. This would be consistent with Figure 1, which shows the catalyst breaking away from the nanoparticle-borohydride complex to start the series of reactions again [47]. However, these reusability trials indicate that while the nanoparticles greatly increase the catalytic activity of the developed catalysts, the AuNP-MCM catalyst is generally less stable.

## 3. Experimental Section

### 3.1. Synthesis

The Mesoporous carbon was reported to be synthesized from starch via the Starbon synthesis method [21,22,24]. Nitrogen sorption/desorption isotherms generated at 77 K for mesoporous carbon is presented in Appendix A and nitrogen adsorption data for mesoporous carbon is included in Appendix A. Gold nanoparticles were synthesized by bringing a 1 mM aqueous chlorauric acid (Sigma Aldrich, St. Louis, MO, USA) solution to a boil and adding 1% aqueous sodium citrate (Sigma Aldrich) dropwise for five minutes with continuous stirring with a magnetic stir bar [48]. The AuNP-MCM nanocomposite was synthesized by adding 40 mL of the nanoparticle solution to 1 g of mesoporous carbon in order to functionalize the mesoporous carbon via the incipient wetness impregnation method [4,5]. The resulting precipitate was filtered and dried in a vacuum oven at 100 °C for 24 h.

### 3.2. Characterization

Transmission electron microscopy (TEM, JEM-2100F, JEOL, Akishima, Tokyo, Japan) was used to visualize the size of the nanoparticles in the composite and to characterize the binding between the nanoparticles and the mesoporous carbon. These samples were prepared by putting 1 μL of the nanoparticle solution onto the TEM sample grid and letting it dry overnight. X-ray diffraction (XRD, Rigaku Miniflex II, Cu Kα X-ray, nickel filters, Rigaku, Tokyo, Japan) was used to determine the crystal structure of mesoporous carbon and its composite. The sample was put onto a P-XRD slide, and a Rigaku Miniflex II was used to perform the P-XRD. Scanning electron microscopy (SEM, JEOL JSM-6060LV, JEOL, Akishima, Tokyo, Japan)/energy dispersive spectroscopy (EDS, ThermoScientific UltraDry, Thermo Fischer Scientific, Waltham, Massachusetts, USA) was used to determine the elemental composition of the nanocomposite and further confirm the presence of nanoparticles on the mesoporous carbon. The sample, in powder form, was mounted on a sample holder with carbon tape and analyzed under SEM with an EDS attachment.

### 3.3. Catalytic Tests

The catalytic properties of the nanocomposite were tested in the hydrogen evolution reaction between water and sodium borohydride (NaBH_4_) as the reducing agent. The volume of generated hydrogen was determined via a gravimetric water displacement system [4,5]. Various conditions, such the concentration of NaBH_4_ (793 µmol, 952 µmol, 1057 µmol), pH (6, 7, 8) and temperature (273 K, 288 K, 295 K, 303 K), were applied in the tests. In all trials, 0.01 g of the AuNP-MCM nanocomposite catalyst was used, and 100 mL of deionized water was used in all NaBH_4_ solutions. During the reaction, the solution was stirred with a magnetic stir bar to maintain the dispersion of the AuNP-MCM. The water displaced during the reaction was measured via an Ohaus Pioneer Balance (Pa124) with proprietary mass logging software. Each variation was repeated in triplicate with the averages calculated. 

### 3.4. Catalyst Reusability

In order to test for the reusability of the AuNP-MCM nanocomposites, a 952 µmol solution of NaBH_4_ and 100 mL of deionized water at pH 7 and 295 K was made and 0.01 g of the nanocomposite was added. The same solution containing the deionized water and the catalyst was used in five reduction reactions, adding a constant amount of NaBH_4_ for each trial. 

## 4. Conclusions

The structure and composition of the AuNP-MCM catalyst was confirmed via transmission electron microscopy (TEM), scanning electron microscopy/energy dispersive spectroscopy (SEM/EDS), and X-ray diffraction (P-XRD). The catalytic activity for this catalyst saw its catalytic activities increase only with increasing NaBH_4_ concentrations, increasing temperature, and lower pH. The variations in temperature allowed for the determination of an activation energy of 30.0 kJ mol^−1^, which, when compared to previously tested inorganic catalysts, represents one of the lower activation energies, making it favorable. Catalytic reusability tests of this catalyst showed that the AuNP-MCM catalyst is stable and produces a consistent volume of hydrogen after five consecutive uses. This stability and low activation energy make this catalyst a competitive option for the hydrolysis of NBH_4_.

## Data Availability

The raw data supporting the conclusions of this article will be made available by the authors on request.

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
