# Peer review of "Gold Nanoparticle Mesoporous Carbon Composite as Catalyst for Hydrogen Evolution Reaction"

_molecules, 2024, doi:10.3390/molecules29153707_

Round 1

Reviewer 1 Report

Comments and Suggestions for Authors

In this work, the authors prepare gold nanoparticle mesoporous carbon composite as catalyst for efficient hydrogen evolution reaction. Various characterizations are applied to analyze this material. However, some important issues must be addressed before acceptance.

1.     The spacing should be added between values and units, e.g., ’5 nm’. The scale bars in Figures 1 and 2 are too small.

2.     To highlight the importance of hydrogen production and improve the readability of the manuscript, the authors should combine some data in the introduction part. Please refer to 10.1039/D3EE02695G for this point.

3.     Obvious shift of XRD peaks is observed in Figure 3, which can be ascribed to the existence of structural strain. Please analyze the possible structural strain by referring to 10.1016/j.jechem.2023.03.033.

4.     To directly prove the existence of Au nanoparticles, EDX mapping should be given.

5.     To analyze the specific C species in this material, Raman is suggested.   

Comments on the Quality of English Language

 Minor editing of English language required

Reviewer 2 Report

Comments and Suggestions for Authors

In this work authors report the synthesis, the characterization of gold nanoparticles supported on mesoporous carbon as a catalyst for the evolution of hydrogen from NaBH4. This study might be interesting, however a similar study, using activated carbon and glass fiber was published from one of the authors in 2020, presenting the same catalytic scheme, the same dependencies and the same conclusions. The unique difference in the paper submitted to Molecules is the use of a mesoporous carbon. However, it is well known that an activated carbon is a typical mesoporous material. Therefore, I do not think that this paper presents novelty and originality.

Moreover, there are some other remarks:

- In introduction the use of gold instead of other metals is not justified

-In Experimental the gas used during the last step of drying at 100°C, is not specified, but when using a carbon it is essential  to specify it.

- Authors claim that their carbon is a mesoporous one, but do not show any BET isotherm/Pores distribution

- They report an hypothesis of reaction scheme, but no supporting results.

Reviewer 3 Report

Comments and Suggestions for Authors

This manuscript can be read quite smoothly with interesting results, but there are still a few parts as listed below that can be adjusted to make it easier for readers to understand with scientific value. Current decision is “minor revision”

1.      As shown in Fig.4, the results show that the higher concentration makes higher performance in generating hydrogen, but authors didn’t mention what will happen if the concentration is higher than 0.00106 mol. Is there any restriction should be written in this paper to improve the rigor and comprehensiveness of research?

2.      As shown in Fig.5, the results show that the lower pH value makes higher performance in generating hydrogen, but authors are suggested to present what will happen if the pH value is lower than 6. Is there any restriction should be written in this paper to improve the rigor and comprehensiveness of research?

3.      As shown in Fig.6, the results show that the higher temperature makes higher performance in generating hydrogen. It can be suggested to mention what will happen if the temperature is higher than 303K. Is there any restriction should be written in this paper to improve the rigor and comprehensiveness of research?

4.      Figure 7 illustrates the relationship between the natural logarithm of the reaction rate constant and the reciprocal of temperature at different temperatures, and from this, the activation energy of the reaction is calculated. Although the data points in the figure show good linearity, the lack of standard deviation information makes it impossible to fully assess the reliability of these data. It is recommended to provide the standard deviation for each data point, which will significantly enhance the rigor and credibility of the results in Figure 7.

5.      Is there any other size of gold nanoparticles used in the research? How to make sure the 20 nm gold nanoparticles are the best conditions?

Reviewer 4 Report

Comments and Suggestions for Authors

Submission ID: molecules-3051120

Title: Gold Nanoparticle Mesoporous Carbon Composite as Catalyst for Hydrogen Evolution Reaction

Dear Editor,

In this study, the authors investigated depositing gold nanoparticles onto mesoporous carbon to form a nano-composite catalyst (AuNP-MCM). The resulting material was then characterized using Transmission Electron Microscopy (TEM), Powder X-Ray Diffraction (P-XRD), and Scanning Electron Microscopy/Energy Dispersive X-Ray Spectroscopy (SEM/EDS). The discussion is rigorous, making the results potentially interesting to researchers in hydrogen generation techniques.  The catalyst's stability and competitively low activation energy suggest promise for hydrogen generation.

I therefore recommend that it be published without any change in results.
